# Beyond Sequential Reranking: Reranker-Guided Search Improves Reasoning Intensive Retrieval

**Haike Xu**
Massachusetts Institute of Technology
`haikexu@mit.edu`

**Tong Chen**
University of Washington
`chentong@cs.washington.edu`

## Abstract

The widely used retrieve-and-rerank pipeline faces two critical limitations: they are constrained by the initial retrieval quality of the top-k documents, and the growing computational demands of LLM-based rerankers restrict the number of documents that can be effectively processed. We introduce Reranker-Guided-Search (RGS), a novel approach that bypasses these limitations by directly retrieving documents according to reranker preferences rather than following the traditional sequential reranking method. Our method uses a greedy search on proximity graphs generated by approximate nearest neighbor algorithms, strategically prioritizing promising documents for reranking based on document similarity. Experimental results demonstrate substantial performance improvements across multiple benchmarks: 3.5 points on BRIGHT, 2.9 on FollowIR, and 5.1 on M-BEIR, all within a constrained reranker budget of 100 documents. Our analysis suggests that, given a fixed pair of embedding and reranker models, strategically selecting documents to rerank can significantly improve retrieval accuracy under limited reranker budget[1].

## 1 Introduction

Retrieval involving complex query-document relationships has recently received significant research attention across scientific reasoning, instruction following, and multi-modal contexts (Su et al. (2024); Weller et al. (2024); Wei et al. (2024); Oh et al. (2024)). While embedding-based retrieval methods Gao et al. (2021); Karpukhin et al. (2020); Formal et al. (2021); Xiao et al. (2023); Meng et al. (2024) enable efficient search over large corpora, they often struggle with complex tasks due to inherent capacity limitations. The retrieve-then-rerank pipeline Nogueira & Cho (2019) addresses this by using computationally intensive rerankers to assess the top-k search results, with recent large language model (LLM)-based rerankers Sun et al. (2023); Pradeep et al. (2023); Qin et al. (2023); Weller et al. (2025); Shao et al. (2025) substantially improving relevance rankings by jointly analyzing query-document pairs.

While LLM-based rerankers greatly improve relevance ranking, their high computational cost limits the number of documents that can be reranked. As a result, accuracy is bottlenecked by the initial retrieval stage. This leads to a key question: *Given a fixed reranker budget, how can we select documents for reranking to maximize accuracy?* In this work, we move beyond the common sequential scan of retrieval results and propose selecting documents for reranking using a search algorithm over document-document similarity.

Inspired by the recent popular graph-based algorithm for nearest neighbor search in the bi-metric setting Xu et al. (2024), we propose Reranker-Guided-Search to bypass sequential reranking. Based on the clustering hypothesis Jardine & van Rijsbergen (1971) that similar passages have similar relevancy to the same query, we run a greedy search on the proximity graph built from Approximate Nearest Neighbor Search (ANNS) algorithm Jayaram Subramanya et al. (2019); Malkov & Yashunin (2018); Fu et al. (2019) on the document embedding. Specifically, given a search query, we first retrieve a list of seed documents close to the query in the embedding space and rerank them. After

---

[1]Our code is available at `https://github.com/xuhaike/Reranker-Guided-Search`

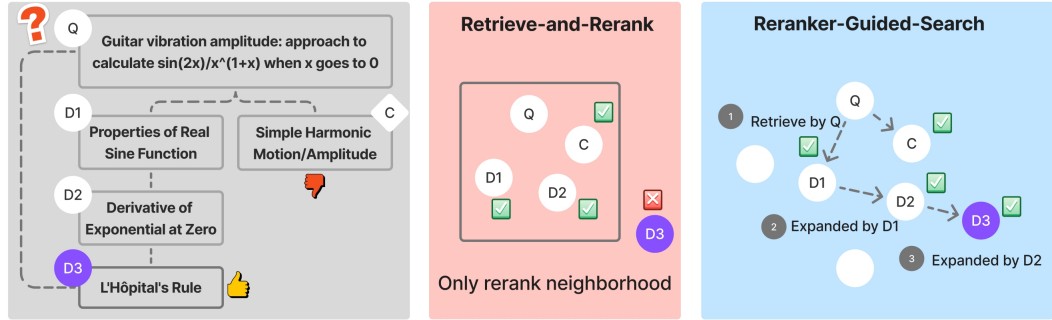

Figure 1: An example from TheoremQA-T illustrating how our Reranker-Guided-Search works. We show the summarized version of the query and document titles on the left.

that, we expand the document candidates by including the neighborhood of those documents favored by the reranker, and rerank those documents again. We iteratively repeat the expand and rerank process until we reach the reranking budget.

Our method enhances retrieval accuracy by strategically allocating the reranking budget. We first rerank a small neighborhood of the query, then avoid spending resources on low-ranked neighborhoods. As illustrated in Figure 1, while query Q and document C1/D1 share surface-level similarities (both Q and D1 mention sine functions, Q and C1 mention amplitude), the reranker assigns C1 a lower rank because it fails to provide the correct mathematical relationship. It then expands D1 to get D2, and D2 to get D3, because document D2, despite minimal word overlap with Q, contains the appropriate method to solve the problem—a quality captured by the reranker. Consequently, our approach finds the correct document D3 by exploring D2's neighborhood while reranking fewer than 500 documents. RGS saves reranker budget by skipping documents similar to the lower-ranked C1. In this case, document D3 was initially ranked 2812th in the embedding space and wouldn't have been found using sequential reranking.

We empirically test our RGS algorithm on three different benchmarks: a reasoning intensive benchmark BRIGHT Su et al. (2024), a multi-modality benchmark M-BEIRWei et al. (2024), and an instruction following benchmark FollowIR Weller et al. (2024). When the reranker budget is fixed to at most 100 documents, our RGS method improves the NDCG@10 score from 25.3 to 28.8 on BRIGHT, from 60.4 to 63.3 on FollowIR, and from 25.9 to 31.0 on M-BEIR, compared to the standard sequential reranking method.

Our analysis shows that in the regime of high reranker budget (e.g. 500), RGS exhibits a unique property that the final retrieval accuracy is less dependent on the embedding model capacity but more decided by the alignment between the reranker model and the groundtruth label. Furthermore, our RGS is robust to query embedding perturbance. It still manages to get reasonable retrieval accuracy even if the similarity between document embedding and query embedding provides little information about their relevancy, which makes RGS suitable for reasoning-intensive retrieval tasks.

In summary, this work makes the following contributions:

- We propose a novel retrieval pipeline RGS, which leverages the similarity of documents to prioritize reranking of more promising documents and improve performance in retrieving complex query-document relationships.

- We propose a rarely discussed evaluation setup—Reranker@k—to assess the efficiency of different methods in applying rerankers and to evaluate their performance on complex query-document matching tasks, including BRIGHT, FollowIR, and M-BEIR.

- We show that, given a pair of embedding and reranker models, strategically selecting documents for reranking can significantly improve accuracy within a fixed reranker budget.

## 2 RELATED WORK

**Graph-based reranking** Recently, several graph-based reranking methods have been proposed to bypass the sequential reranking bottleneck. GAR MacAvaney et al. (2022) builds a KNN graph over the corpus and maintains a graph frontier of reranked documents. It alternates between reranking documents from the shortlist returned by vector similarity search and from the graph frontier. RAR Frayling et al. (2024) extends GAR by incorporating the query point and constructing a bipartite graph between documents and queries. SlideGAR (Rathee et al., 2025) further extends GAR to accommodate listwise rerankers. From a practical perspective, all prior works (GAR, RAR, SlideGAR) evaluate their methods only on the classical TREC dataset (Craswell et al., 2020), which does not capture the complexity of modern user queries. In contrast, our analysis shows that RGS exhibits strong performance on more complex retrieval tasks.

**Graph-based ANNS algorithms** Graph-based ANN algorithms Jayaram Subramanya et al. (2023); Malkov & Yashunin (2018); Fu et al. (2019) are a family of heuristic-based methods that have recently gained increased attention in the ANN community. These algorithms typically build a proximity graph over high-dimensional vectors and then perform a greedy search to traverse toward the query's nearest neighbors. Although Indyk & Xu (2023) has shown that graph-based methods lack worst-case theoretical guarantees, they have demonstrated strong empirical performance. In addition to excelling at standard ANN tasks, graph-based algorithms offer other advantages. For example, Xu et al. (2024) proposes a bi-metric framework that leverages a proximity graph built using the DiskANN algorithm under one distance metric to search for nearest neighbors according to a different distance metric. This idea inspired us to design Reranker-Guided-Search for reranking.

## 3 METHOD

### 3.1 PRELIMINARIES

Given a document corpus $C = \{d_1, ..., d_n\}$ and a query $q$, we have access to an embedding model $E(\cdot)$ that maps queries and documents to a shared embedding space, and a reranker model $D$ that evaluates the relevance between a query and a document. An ideal embedding model $E$ should satisfy two conditions:

- document-wise similarity: Given two documents $d_1$ and $d_2$, the more similar they are, the larger the inner product between $E(d_1)$ and $E(d_2)$.
- query-document relevance: Given a query $q$ and a document $d$, the more relevant the document is to the query, the larger the inner product between $E(q)$ and $E(d)$.

The goal of retrieval task is to return the most relevant document to the query in the most efficient way. In the scope of our paper, we focus on how to better utilize the reranker model. Our objective is to find the most relevant document while under fixed reranker calls budget.

### 3.2 BASELINE

We mostly compare our RGS method with two other baselines: Retrieve-and-Rerank and SlideGAR. Here, we briefly describe SlideGAR, introduced in a recent work Rathee et al. (2025). SlideGAR performs reranking in rounds until its reranker call quota is exhausted. In each round, it asks the reranker to rerank $w$ documents and retains the top $w/2$ among them. The $w$ candidates are composed of two parts: one half comes from the top $w/2$ documents selected in the previous round, and the other half is alternately sampled from either the initial short list sorted by vector similarity or from the KNN graph neighbors of the $w$ documents reranked in the previous round.

Although both SlideGAR and our RGS explores the graph frontier on the corpus graph, we highlight key algorithmic differences between them and our RGS method. First, our RGS maintains an up-to-date document order by the reranker's preference. RGS explores all the neighbors of the most promising document and inserts them to the sorted list. In contrast, SlideGAR only picks $w/2$ documents from the graph frontier with the top vector similarity score, discarding the rest of them. This means that even if a relevant document appears in the graph frontier, it may not be ranked

immediately if its embedding is not close to the query. Thus, we view SlideGAR more like an augmented linear rerank with the addition of the graph neighbors from the current candidates, while RGS tries to simulate a standard beam search used in graph-based approximate nearest neighbor search algorithms.

### 3.3 Reranker-Guided-Search

Inspired by the clustering hypothesis Jardine & van Rijsbergen (1971), we believe that if a document is judged to be relevant by a reranker, then its similar documents are likely to be relevant as well. We thus prioritize to rerank those documents whose similar documents have been assigned high scores by the reranker. We combine this idea with the bi-metric techniques from Xu et al. (2024) to design Reranker-Guided-Search.

Our RGS algorithm first builds a proximity graph over documents by running the DiskANN algorithm Jayaram Subramanya et al. (2019) on the document embeddings. DiskANN incrementally constructs the proximity graph by inserting new points into the graph data structure. For each new point $p$, it first performs a greedy search on $p$ to obtain the set of visited points $V$. Then it creates bi-directional edges between $p$ and points in $V$. Finally, a robust pruning procedure is applied to any points with more than $R$ edges. After preprocessing, DiskANN produces a proximity graph in which each point $v$ has at most $R$ outgoing neighbors $N_{out}(v)$. (Please refer to Jayaram Subramanya et al. (2019) for details)

Given a query $q$, RGS performs a two-stage search to locate the relevant document. In stage one, we run DiskANN to find the document $s$ whose embedding is closest to that of the query. In stage two, we run a greedy search guided by the reranker, starting from document $s$. Specifically, RGS maintains an ordered list $A$ of the most relevant documents found so far. Initially, the list only contains one document $s$. At each step, we pick the first unexpanded document $v$ in the list, explore its neighbor documents $N_{out}(v)$ in the graph, and append them (at most $R$ documents) to the end of the list. We then use a listwise reranker to reorder the newly added documents from the end toward the front using a sliding window. This expand and rerank process is repeated until we reach the reranker call budget. Finally, the first document in list $A$ is returned as the most relevant document to the query. Please refer to Algorithm 1 for details.

In our implementation, we focus on a listwise reranker model $D$, as Sun et al. (2023) suggests it performs better than a pointwise reranker when applied to LLMs. Our RGS algorithm can be easily modified to accommodate a pointwise reranker by replacing the list $A$ with a priority queue sorted by reranker scores.

---

**Algorithm 1** Reranker-Guided-Search($q$)

1: **Input**: Graph index $G$, listwise-reranker $D$, query $q$, search list size $L$, sliding window size $w$.
2: **Output**: the most relevant document for query $q$
3: $s \leftarrow$ Use DiskANN algorithm to search for the closest vector from query embedding
4: $A \leftarrow \{s\}$
5: $U \leftarrow \varnothing$
6: **while** $A \setminus U \neq \varnothing$ **do**
7:     $v \leftarrow$ the first vertex in $A \setminus U$
8:     $U \leftarrow U \cup v$
9:     Append $N_{out}(v) \setminus A$ to the end of $A$            $\triangleright N_{out}(v)$: out neighbors of node $v$ in $G$
10:     **for** $i = |A|$ down to 0 step size $w/2$ **do**
11:         Use reranker D to reorder A[i-w $\cdots$ i]
12:     **if** $|A| > L$ **then**
13:         $A \leftarrow$ the first $L$ vertices in A
14: **return** the first element in $A$

---

### 3.4 Analysis

We first analyze the running time of our RGS algorithm. Since the time bottleneck lies in calling the reranker model, we focus on bounding the number of reranker calls. Empirically, the number of expanded nodes is usually bounded by $O(L)$, where $L$ is the search list size. For each expansion,

we run the reranker $D$ to reorder the search list using a sliding window with step size $w/2$, which involves $O(L/w)$ reranker calls. Overall, the estimated number of reranker calls is $O\left(\frac{L^2}{w}\right)$.

For retrieve-and-rerank and SlideGAR, since they are budget-based algorithms, suppose the reranker budget is $Q$ and the window size is $w$. Then the estimated number of reranker calls is $O(Q/w)$.

Regarding accuracy, we can invoke Theorem 1.1 from Xu et al. (2024), which shows that as long as the vector similarity approximates the reranker score (e.g., for a pointwise reranker) within a constant factor, our RGS algorithm is able to find the most relevant documents according to the reranker score.

## 4 EXPERIMENTS

To evaluate the performance of our RGS method, we compare RGS with other two methods (Retrieve-and-Rerank, and SlideGAR (Rathee et al., 2025)) and test them on three benchmarks: BRIGHT (Su et al., 2024) (reasoning intensive), M-BEIR (Wei et al., 2024) (multi-modality), and FollowIR (Weller et al., 2024) (instruction following). We believe that these benchmarks represent the most complicated retrieval tasks available, which can't be solved by simple semantic similarity, and serve as a better testbed for evaluating the efficiency of different high-performing retrieval methods.

### 4.1 EXPERIMENTAL SETUP

**Dataset Details**   BRIGHT Su et al. (2024) is a reasoning-intensive benchmark composed of naturally occurring human data from 12 different domains. Relevance in this dataset goes beyond simple lexical or semantic similarity; the authors constructed hard negatives that are topically related to the query but do not meet its specific requirements. M-BEIR Wei et al. (2024) is a multimodal retrieval benchmark built from eight different query-corpus modality subtask. For each subtask, the query can be a text $q_t$, an image $q_i$, or a combination of both. Each candidate can be a text $c_t$, an image $c_i$, or a combination $(c_i, c_t)$. It poses a challenge for retrieval methods, as models must integrate knowledge from multiple modalities to interpret user queries and return answers in the specified modality. FollowIR Weller et al. (2024) is an instruction-following retrieval benchmark curated from three TREC datasets, in which annotators create different instructions for each query to test a model's ability to adjust retrieval results based on nuanced differences in user intent. We believe these three benchmarks all demand a non-trivial degree of reasoning compared to classical IR tasks and are therefore more suitable for evaluating the capabilities of retrieval methods to capture complicated query-document relationships.

**Metrics**   We propose a rarely discussed evaluation setup Reranker@k, which limits the reranker to scan at most $k$ documents. In our experiment, we set $k = 100/300/500$ and measure NDCG@10 under different reranker budget. Note that our setup allows each document to be reranked multiple times by a listwise reranker. We also consider the number of token usage, API calls in Appendix A. For M-BEIR, we randomly sample a subset of 100 queries to test and we remove those groundtruth labels not appearing in the corpus for accurate NDCG@10 evaluation.

**Baselines and Algorithm Details**   We compare our method with two baselines shown as follows:

- Retrieve-and-Rerank: For different reranker budget $k = 100/300/500$, it uses DiskANN algorithm to retrieve the top $k$ documents closest to the query embedding and rerank them in a sliding-window way Sun et al. (2023) with window size $w = 10$.

- SlideGAR: We reimplement the algorithm in Rathee et al. (2025). First, SlideGAR uses the DiskANN algorithm to retrieve a shortlist of the top $k$ documents closest to the query embedding. Next, it builds a KNN graph based on document-document similarity. It then alternates between reranking documents from either the retrieved shortlist or the graph frontier until the reranker budget is reached. Following Rathee et al. (2025), we set the window size to $w = 20$.

- Reranker-Guided-Search: In practice, we make the following adaptions to Algorithm 1. The search list size $Ls$ is tuned based on the reranker budget $k$ ($Ls = 20/30/50$ for $k = 100/300/500$). We initialize our search with $k/5$ start points returned by the first stage vector similarity search. The algorithm is forced to immediately return the best documents seen so far once the reranker budget $k$

is exhausted We set window size to $w = 10$ for our RGS method and return the top-10 documents in list $A$ for evaluating NDCG@10.

**Embedding/Reranker Model Details**  We choose BGE-Large Xiao et al. (2023) as the embedding model for BRIGHT and FollowIR, and CLIP Radford et al. (2021) embeddings for M-BEIR. We prompt Gemini-2.0-Flash to be our reranker model. Please refer to Appendix A for our prompt. For the multi-modality benchmark M-BEIR, when either the query or a candidate in the corpus contains both image and text information, we use CLIP's text and image encoders to separately encode each modality, and then apply element-wise addition to produce the final embedding.

## 4.2  RESULTS

**Scientific and Mathmatical Reasoning**  The evaluation results on BRIGHT are presented in Table 1. Using dense retrieval with `BGE-Large` as the encoder yields an average NDCG@10 of 13.7 across all datasets. When applying a retrieval-and-rerank pipeline, sequential reranking of the top-100 documents improves the score to 25.3, and scanning the top-500 documents yields 27.7. This suggests that dense retrieval alone struggles to capture complex query-document relationships, which rerankers can help improve.

SlideGAR achieves 25.4 at Reranker@100 and 26.9 at Reranker@500, showing minimal improvement over sequential scanning. In contrast, our method, RGS, achieves 28.8 (+14%) at Reranker@100 and 33.0 (+19%) at Reranker@500, demonstrating a substantial improvement in accuracy under the same reranker budget.

We also observe that RGS provides the largest gains over sequential scanning in scientific reasoning tasks, whereas the improvement is minimal on AoPS and LeetCode. We speculate that reranking a list of math or code examples requires intensive reasoning ability, which remains challenging—even when using Gemini-Flash-2.0 as the reranker. In these cases, no reranking method is able to improve upon the first-stage retrieval.

**Instruction Following**  The evaluation results on FollowIR are presented in Table 2. Using dense retrieval with `BGE-Large` as the encoder yields an average NDCG@10 of 49.9 across all datasets. Applying a retrieve-and-rerank pipeline, sequentially reranking the top-100 documents improves the score to 60.4, while scanning the top-500 documents yields 61.6. This suggests that sequential reranking provides limited benefit in capturing user instructions.

SlideGAR achieves 62.5 at Reranker@100 and 59.8 at Reranker@500, showing a slight decrease as the reranker budget increases. In contrast, our method, RGS, achieves 63.3 (+5%) at Reranker@100 and 64.2 (+4%) at Reranker@500, demonstrating a stable improvement in accuracy under the same reranker budget.

Compared with the results on BRIGHT, the performance gap between different methods is smaller on FollowIR. We observe that on FollowIR, over 87% of the ground-truth answers are located within the top 100 documents ranked by query-document vector similarity, whereas the number is only 31% on BRIGHT. This indicates that the marginal benefit of searching beyond the top 100 is limited. Since most of the newly discovered documents beyond the top 100 are likely to be noise—and the LLM-based reranker doesn't always align with ground-truth labels-this explains the slight performance drop observed in some methods as the reranker budget increases.

**Multi-modality retrieval**  The evaluation results on M-BEIR are presented in Table 3. Using dense retrieval with `clip-ViT-B-32` as the encoder yields an average NDCG@10 of 15.3 across all datasets. When applying a retrieval-and-rerank pipeline, sequential reranking of the top-100 documents improves the score to 25.9, and scanning the top-500 documents yields 30.1. This suggests that dense retrieval alone struggles to capture cross-modality retrieval tasks, which rerankers can help improve.

SlideGAR achieves 28.6 at Reranker@100 and 29.9 at Reranker@500, showing minimal improvement over sequential scanning. In contrast, our method, RGS, achieves 31.0 (+20%) at Reranker@100 and 38.3 (+27%) at Reranker@500, demonstrating a substantial improvement in accuracy under the same reranker budget.

We notice that the improvement is significant on certain tasks. For example, on task 6, while the standard CLIP embedding scored less than 5 even after reranking the top-500 candidates, our RGS achieves an NDCG@10 score of 30.7 and 15.6 using the same reranker budget. In Wei et al. (2024), the authors mention that they use the 'Wikipedia page title' as text input rather than using the '100 tokens from the Wikipedia page' as the candidate because it provides better 'zero-shot performance.' This indicates that they are aware of the limited ability of embedding models to encode long context, which we believe is an ideal case to demonstrate the advantage of our reranking method.

| Dataset | No reranker BGE-Large | Reranker@100 | | | Reranker@300 | | | Reranker@500 | | |
|---|---|---|---|---|---|---|---|---|---|---|
| | | RR | SlideGAR | RGS | RR | SlideGAR | RGS | RR | SlideGAR | RGS |
| Biology | 11.7 | 31.1 | 33.9 | **37.0** | 37.6 | 37.0 | **42.8** | 38.9 | 40.7 | **46.8** |
| Earth Science | 24.6 | 42.2 | 43.7 | **45.3** | 43.0 | 43.2 | **47.1** | 38.5 | 43.5 | **49.2** |
| Economics | 16.6 | 23.0 | 26.0 | **26.2** | 27.4 | **29.1** | 28.5 | 28.8 | 28.8 | **30.4** |
| Psychology | 17.5 | 32.0 | 33.1 | **40.9** | 39.5 | 38.5 | **45.4** | 40.4 | 41.1 | **49.2** |
| Robotics | 11.7 | 24.5 | 24.6 | **26.5** | 22.9 | 28.0 | **29.5** | 25.0 | 21.9 | **29.8** |
| Stack Overflow | 10.6 | 27.2 | 26.2 | **29.9** | 27.5 | 26.7 | **29.1** | 27.5 | 24.2 | **27.7** |
| Sustainable Living | 13.1 | 30.8 | 31.4 | **35.5** | 24.7 | 35.5 | **36.2** | 26.8 | 30.8 | **37.1** |
| LeetCode | 26.7 | **26.0** | 17.4 | 25.3 | **24.5** | 17.0 | 23.5 | **25.8** | 11.2 | 22.1 |
| Pony | 5.7 | **24.5** | 21.6 | 22.5 | 21.1 | 19.8 | **21.9** | 23.7 | 20.8 | **25.1** |
| AoPS | 6.0 | 6.7 | 6.9 | **7.1** | 7.5 | 4.0 | **7.8** | 6.7 | 3.5 | **7.4** |
| TheoremQA-Q | 13.0 | 20.1 | 17.7 | **21.2** | 22.3 | 20.6 | **28.6** | 25.8 | 23.7 | **30.2** |
| TheoremQA-T | 6.9 | 15.8 | 21.9 | **28.2** | 20.8 | 29.6 | **39.3** | 24.5 | 33.0 | **40.7** |
| Avg | 13.7 | 25.3 | 25.4 | **28.8** | 26.6 | 27.4 | **31.6** | 27.7 | 26.9 | **33.0** |

Table 1: BRIGHT evaluation with varying reranker budgets for RR (retrieve-and-rerank), SlideGAR (Rathee et al., 2025), and our RGS (Reranker-Guided-Search).

| Dataset | No reranker BGE-Large | Reranker@100 | | | Reranker@300 | | | Reranker@500 | | |
|---|---|---|---|---|---|---|---|---|---|---|
| | | RR | SlideGAR | RGS | RR | SlideGAR | RGS | RR | SlideGAR | RGS |
| Robust04 | 47.2 | 58.7 | 59.2 | **61.6** | 59.6 | 56.6 | **63.3** | 61.0 | 55.9 | **65.0** |
| News21 | 52.6 | 57.4 | 58.5 | **58.8** | **58.7** | 57.3 | 58.6 | **58.6** | 56.5 | 58.2 |
| Core17 | 49.8 | 65.0 | **69.7** | 69.6 | 64.3 | 66.7 | **70.3** | 65.2 | 67.1 | **69.5** |
| Avg | 49.9 | 60.4 | 62.5 | **63.3** | 60.9 | 60.2 | **64.1** | 61.6 | 59.8 | **64.2** |

Table 2: FollowIR evaluation with varying reranker budgets for RR (retrieve-and-rerank), SlideGAR (Rathee et al., 2025), and our RGS (Reranker-Guided-Search).

| Task | Dataset | No reranker CLIP | Reranker@100 | | | Reranker@300 | | | Reranker@500 | | |
|---|---|---|---|---|---|---|---|---|---|---|---|
| | | | RR | SlideGAR | RGS | RR | SlideGAR | RGS | RR | SlideGAR | RGS |
| 1. $q_t \to c_i$ | VisualNews | 24.4 | 36.4 | 36.9 | **37.4** | 35.1 | 39.8 | **39.9** | 35.8 | **41.2** | 36.5 |
| | MSCOCO | 41.1 | **59.4** | 55.6 | 55.8 | **58.7** | 55.3 | 57.6 | **60.3** | 54.6 | 55.1 |
| | Fashion200K | 2.4 | **7.4** | 5.9 | 5.6 | **10.3** | 6.8 | 8.6 | 8.9 | **9.9** | 7.7 |
| 2. $q_t \to c_t$ | WebQA | 22.7 | 44.3 | 49.8 | **52.7** | 48.4 | 53.6 | **62.8** | 50.3 | 54.3 | **67.9** |
| 3. $q_t \to (c_i, c_t)$ | EDIS | 24.5 | 43.2 | 46.6 | **52.6** | 45.2 | 46.7 | **53.6** | 49.0 | 48.2 | **55.3** |
| | WebQA | 20.5 | 42.8 | 46.4 | **48.4** | 45.7 | 45.5 | **53.6** | 48.6 | 46.4 | **54.3** |
| 4. $q_i \to c_t$ | VisualNews | 24.0 | 30.4 | **33.3** | 27.3 | 31.1 | 30.9 | **32.3** | 30.3 | 29.5 | **33.3** |
| | MSCOCO | 35.8 | 44.3 | 44.5 | **46.5** | 44.5 | 44.6 | **44.7** | 44.9 | 43.8 | **45.3** |
| | Fashion200K | 1.6 | 4.7 | 5.0 | **5.3** | 5.9 | 8.0 | **9.5** | 9.2 | 6.9 | 8.7 |
| 5. $q_i \to c_t$ | NIGHTS | 24.4 | 26.7 | 24.0 | **28.9** | **29.8** | 26.0 | 26.0 | **31.0** | 26.2 | 25.4 |
| 6. $q_i \to (q_t, c_t)$ | OVEN | 0.0 | 0.0 | 2.9 | **4.2** | 3.0 | 3.7 | **21.4** | 3.0 | 5.7 | **30.7** |
| | InfoSeek | 0.1 | 1.0 | 2.1 | **3.9** | 1.3 | 4.1 | **6.8** | 0.7 | 4.2 | **15.6** |
| 7. $q_i \to (q_t, c_i)$ | FashionIQ | 2.7 | 8.1 | **12.2** | 11.8 | 11.6 | 11.0 | **16.0** | 9.8 | 8.8 | **15.3** |
| | CIRR | 12.8 | 31.8 | 33.3 | **33.6** | 40.8 | 35.6 | **47.5** | 41.6 | 34.9 | **49.2** |
| 8. $(q_i, q_t) \to (c_i, c_t)$ | OVEN | 5.4 | 14.8 | 33.2 | **41.4** | 15.7 | 36.8 | **57.9** | 29.9 | 36.8 | **59.5** |
| | InfoSeek | 2.8 | 18.8 | 25.2 | **40.4** | 28.8 | 20.0 | **47.0** | 28.4 | 27.2 | **52.6** |
| | Avg | 15.3 | 25.9 | 28.6 | **31.0** | 28.5 | 29.3 | **36.6** | 30.1 | 29.9 | **38.3** |

Table 3: M-BEIR evaluation with varying reranker budgets for RR (retrieve-and-rerank), SlideGAR (Rathee et al., 2025), and our RGS (Reranker-Guided-Search).

## 5 ANALYSIS

In this section, we investigate how the quality of the embedding and reranker models, as well as noise in the embeddings, affect retrieval accuracy. We perform an error analysis to identify the source of missing ground-truth results and conduct ablation studies on the impact of first-stage retrieval and various graph types.

### 5.1 RETRIEVAL RESULTS WITH DIFFERENT EMBEDDING / RERANKER MODELS

We study the extent to which embedding quality affects final retrieval accuracy. Our experiments are conducted on a selected subset of datasets—Biology, Psychology, Sustainable Livings, and TheoremQA-T from the BRIGHT benchmark. We choose these four datasets because they exhibit the largest performance gaps between our RGS method and RR, making the algorithmic behavior of our method more pronounced and easier to analyze. To examine the impact of embedding quality, we apply four embedding models of varying sizes: `SFR-Mistral` (7B), `BGE-Large` (300M), `BGE-Base` (100M), and `BGE-Micro` (17M).

As shown in Figure 2, for the retrieve-and-rerank method, we observe a consistent performance gap between stronger and weaker embedding models. In cases where the reranker model has significantly higher capacity than the embedding model, the final retrieval accuracy is largely constrained by how many ground-truth answers are retrieved during the first-stage ANN search.

For our method RGS , although more powerful embedding models achieve higher retrieval accuracy at earlier stages, all models converge to similar accuracy under the Reranker@500 setting. This supports our claim that the embedding model primarily determines how many reranker calls are needed to reach a given retrieval accuracy, while the final performance is ultimately bounded by the reranker's capability.

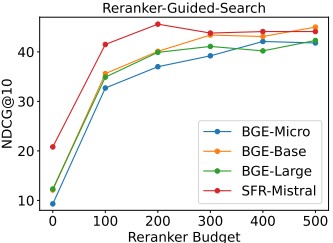 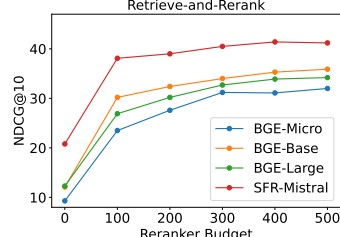

Figure 2: Average NDCG@10 result over a selected subset of datasets from BRIGHT. The left figure is our method RGS, the right figure is retrieve-and-rerank. Experiments involve 4 different embedding models with varying embedding quality.

We also study how our method performs across different reranker models. In Table 5 from Appendix A, we present experimental results on a subset of BRIGHT datasets using "gpt-4.1-mini". We observe that our RGS method consistently achieves the highest retrieval accuracy across various reranker budgets and datasets, although the final accuracy varies compared to Table 1. This demonstrates the generalization ability of our RGS method, and highlights that different reranker models have their own specialties.

### 5.2 DIFFERENT ROLES OF DOCUMENT-DOCUMENT SIMILARITY AND QUERY-DOCUMENT RELEVANCE IN THE RETRIEVAL PROCESS

We perform a fine-grained ablation study to examine how embedding quality affects the retrieval accuracy of the three methods: RR, SlideGAR, and RGS. As mentioned in Section 3, we believe that an ideal embedding should satisfy two conditions: preserving document-wise similarity and approximating query-document relevance. In the following, we investigate the role of each by introducing different levels of perturbation to either the query or document embeddings.

**Perturbation on query embeddings**  Fix a ratio $w \in [0, 1]$, we mix each query embedding with another random query embedding according to the ratio $1 - w : w$. This simulates the situation where the embedding model fails to distinguish subtle differences between queries. In this case, the embedding model's ability to measure query-document relevance is compromised, while the document-wise similarity structure remains intact, as the document embeddings are unchanged.

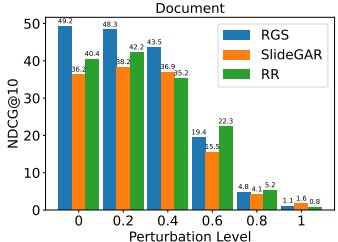 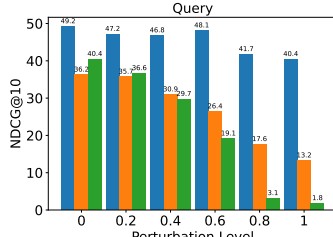

Figure 3: For different levels of perturbation added on query (right) / document(left) embeddings, we report the retrieval accuracy for different retrieval methods

**Perturbation on document embeddings** Fix a ratio $w \in [0,1]$, we mix each document embedding with another random document embedding according to the ratio $1 - w : w$. This simulates the scenario where the embedding model fails to distinguish subtle differences between documents. In this case, both key functions of the embedding model—measuring query-document relevance and preserving document-wise similarity—are compromised.

Please see Figure 3 for retrieval result on the Psychology dataset from BRIGHT after applying two types of perturbations to the query and document embeddings. Note that $w = 0$ corresponds to the original, unaltered embeddings, while $w = 1$ represents the case where the embedding model completely confuses one document/query with another.

We observe that for RR, its performance remains consistent regardless of whether the perturbation is applied to the query or the document, as adding noise to either is symmetric under the sequential reranking process. Its retrieval accuracy drops to nearly zero as $w$ increases from $0$ to $1$.

For SlideGAR, retrieval accuracy degrades more slowly when the perturbation is applied to the query embedding. This is because, even if a completely incorrect query embedding results in a meaningless first-stage shortlist, SlideGAR can still leverage document-document similarity to approach the correct answer.

For our RGS method, retrieval accuracy declines only slightly even when the query embedding contains no useful information (i.e., when $w = 1$). This is because the query embedding is used solely to initialize the starting point for RGS's second-stage search. A random starting point may slow down the search process but has limited impact on the final retrieval result.

## 5.3 ERROR ANALYSIS

We perform an error analysis to understand why the performance of RGS stops improving as the reranker budget continues to grow. We categorize the ground-truth answers into three classes: (1) answers returned by the retrieval algorithm (orange), (2) answers seen during the retrieval process but not selected by the reranker (light orange), and (3) answers not seen at all during the retrieval process (gray). We plot the percentage of each class as the reranker budget increases for both RGS and the retrieve-and-rerank method (see Figure 4). We observe that although the percentage of returned answers does not increase—and sometimes even slightly decreases—the percentage of seen answers consistently grows with the reranker budget. This suggests that the final retrieval accuracy is limited not by the efficiency of our retrieval algorithm, but by the fact that the reranker model's preferences do not always align with human-labeled relevance.

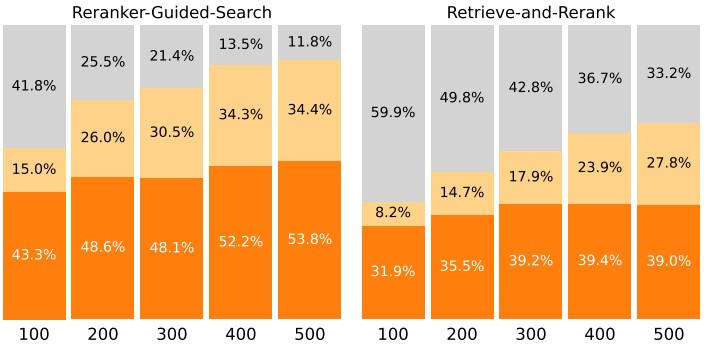

Figure 4: For different Reranker budgets, each stacked bar shows the percentage of ground-truth passages that were retrieved (orange), examined but not selected (light orange), or never examined (grey). Results are reported for running RGS (left) and RR (right) methods on BRIGHT/Biology dataset.

## 5.4 IMPACT OF FIRST STAGE RETRIEVAL

Inspired by the observation that our RGS method still performs reasonably well even when the query embedding is completely incorrect, we question whether the first-stage nearest neighbor search based on the query embedding is necessary. To investigate this, we experiment with different initialization strategies for RGS (see Figure 5). When we initialize RGS with a noisy starting point (e.g., the 1000th closest vector to the query) or use the default start point of the DiskANN algorithm, our method still achieves comparable retrieval accuracy—albeit with increased reranker usage. Notably, the default start point strategy does not require a first-stage search via vector similarity, nor does it require the query embedding at all. This indicates that the initial vector similarity search is not essential for our method. In fact, when the first-stage search is entirely removed, the query embedding becomes unnecessary; all we need from the document embeddings is their encoding of document-wise similarity.

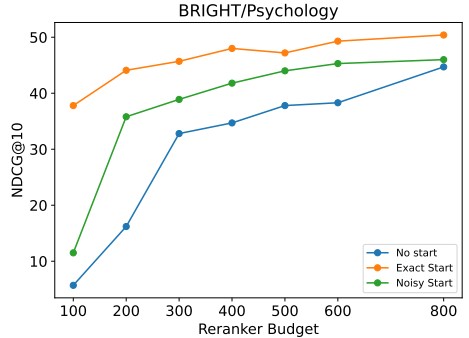 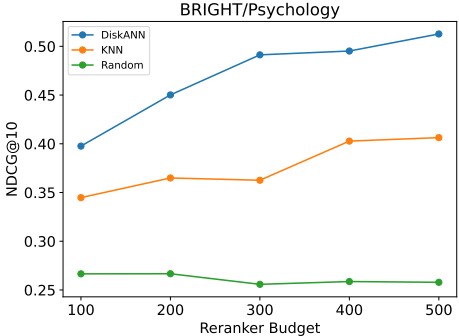

Figure 5: Impact of choosing different start points to initialize our second stage search on retrieval accuracy. Exact start means we start at the closest document to the query in the embedding space. Noisy start means we start at the 1000th closest document to the query. No start means we start at the default start point of the ANNS data structure.

Figure 6: Impact of running our second-stage search on different graph data structures for retrieval accuracy. DiskANN: proximity graph generated by the DiskANN algorithm. KNN: each document is connected to its k nearest neighbors in the embedding space. Random: a randomly connected graph with degree k.

## 5.5 IMPACT OF DIFFERENT GRAPH TYPES

Our RGS method relies heavily on the underlying graph data structure which reflects docuemnt-wise similarity. Here, we investigate whether our RGS works on different graph structures. Besides the proximity graph built by the DiskANN algorithm, we also consider random graphs and KNN graphs, which are common choices in recent graph-based methods in the literature (Rathee et al. (2025); MacAvaney et al. (2022)). Please see Figure 6. We initialize RGS on the 50 documents closest to the query and observe how quickly the greedy search approaches the true answer. RGS still manages to improve on KNN graphs but the process is much slower, and it shows no improvement on random graphs. We hypothesize that in KNN graphs, edges exist only between nearby documents, while connections between distant documents are missing, preventing the search from reaching the correct neighborhood. This also explains why graph-based ANN algorithms (e.g., HNSW, DiskANN, NSG) do not perform greedy search on KNN graphs. For random graphs, it is expected that RGS shows no improvement because the search is equivalent to randomly scanning documents in the corpus.

## 6 CONCLUSION

In this paper, we propose Reranker-Guided-Search, to bypass the sequential reranking bottleneck. Our method leverages a proximity graph built on document-wise similarity and performs a greedy search guided by the reranker to traverse toward the correct answer. We conduct extensive experiments to demonstrate the effectiveness of our method on three reasoning-intensive retrieval benchmarks. Our ablation studies show that the quality of the embedding model affects the reranker budget required to achieve a certain level of retrieval accuracy, while the final retrieval accuracy is primarily determined by the reranker model. A limitation of our RGS method is that adaptively choosing documents to rerank may hinder parallelization, which we leave for future work.

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

## A    FULL EXPERIMENTAL DETAILS AND RESULTS

**Prompt for Gemini-2.0-Flash**    Here, we use a similar prompt from Sun et al. (2023) to ask Gemini-2.0-Flash to rerank the documents.

| System Instruction | You are RankGPT, an intelligent assistant that can rank answers based on their relevance to the query. I will provide you with 10 passages, each indicated by a number identifier []. Rank the answers based on their relevance to query: {query}. |
|---|---|
| Messages | [1] {Passage 1}
[2] {Passage 2}
...
[10] {Passage 10}
Query: {query}. Rank the 10 passages above based on their relevance to the query. The passages should be listed in descending order using identifiers. The most relevant passages should be listed first. The output format should be like [1] >[2] ... >[10]. Only response the ranking results, do not say any word or explain. |

Table 4: Prompt for "Gemini-2.0-Flash" to serve as a reranker

**Computational resource to reproduce our experiments**    Our experiments are run on an Intel(R) Xeon(R) Platinum 8481C CPU with 44 cores and we use one NVDIA A100 GPU to generate the embeddings. The estimated cost spent on Gemini-2.0-Flash to reproduce Table 1, Table 2, and Table 3 is $600.

**More metrics**    Previously, we defined the reranker budget as the number of documents seen by the reranker. However, this may not accurately reflect the total computational resources consumed during retrieval, as each document can be seen multiple times by a listwise reranker. Here, we plot the average number of tokens or API calls sent to the LLM-based reranker per query versus retrieval accuracy. Please see Figure 7 for the results. We observe that RGS achieves the highest retrieval accuracy on BRIGHT under a fixed budget, whether measured by token usage or API calls.

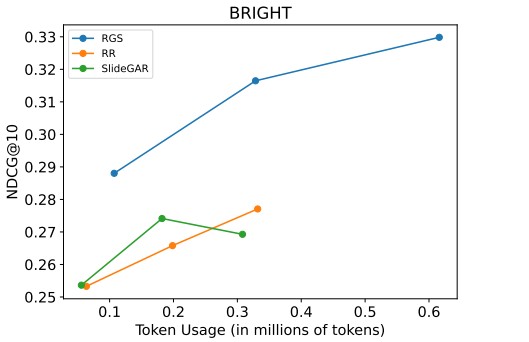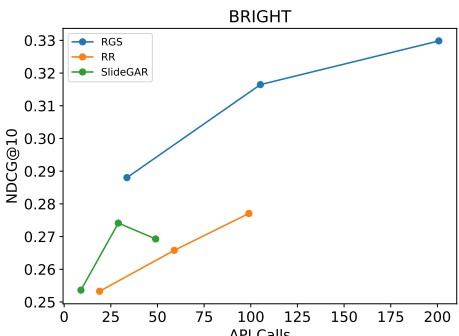

Figure 7: Average NDCG@10 versus number of tokens or API calls sent to the LLM-based reranker on the BRIGHT benchmark

**Experimenting with "gpt-4.1-mini"**    We experiment with "gpt-4.1-mini" on a subset of the BRIGHT datasets. We observe that our RGS method consistently achieves the highest retrieval accuracy across various reranker budgets and datasets, although the final accuracy varies compared to the results in Table 1. This demonstrates the generalization ability of our RGS method and highlights that different reranker models have their own strengths.

**An example from TheoremQA-T**    Below we show an detailed example from TheoremQA-T, where our RGS algorithm is able to traverse the corpus graph and gradually approach the ground-truth

| Dataset | No reranker BGE-Large | Reranker@100 | | | Reranker@300 | | | Reranker@500 | | |
|---|---|---|---|---|---|---|---|---|---|---|
| | | RR | SlideGAR | RGS | RR | SlideGAR | RGS | RR | SlideGAR | RGS |
| Biology | 11.7 | 34.4 | 42.6 | **46.3** | 44.5 | 52.3 | **56.6** | 46.0 | 53.0 | **57.8** |
| Psychology | 17.5 | 32.3 | 34.7 | **42.8** | 43.2 | 42.2 | **50.7** | 41.9 | 42.3 | **49.4** |
| Sustainable Living | 13.1 | 33.3 | 38.5 | **41.1** | 36.7 | 42.3 | **42.9** | 36.7 | 43.2 | **43.5** |
| TheoremQA-T | 6.9 | 18.0 | 24.4 | **29.9** | 26.0 | 35.8 | **42.4** | 31.1 | 38.5 | **41.4** |

Table 5: A subset of the BRIGHT evaluation on "gpt-4.1-mini" with varying reranker budgets for RR (Retrieve-and-Rerank), SlideGAR (Rathee et al., 2025), and our RGS (Reranker-Guided-Search).

answer, even when it is ranked low under cosine similarity. The first row is the query, and the second row is the starting point of our RGS algorithm. Each subsequent row is a neighbor of the previous document in the corpus graph. Following RGS, we eventually reach Document 11592, which is ranked 2812 by cosine similarity but is in fact the ground-truth answer for the query.

# B    USAGE OF LARGE LANGUAGE MODELS

We use LLMs to guide our search algorithm in our experiments. We also use LLMs to polish writing.

| Type | ID | Embedding Rank | Content |
|------|------|------|---------|
| **Query** | – | – | Imagine you're trying to tune a guitar. For each tuning setting $x$, the amplitude is $$\left(\frac{\sin(2x)}{x}\right)^{1+x}.$$ What does the amplitude approach as $x \to 0$? |
| Document | 19940 | 22 | **Properties of the Real Sine Function.** Let $x \in \mathbb{R}$. Properties of $\sin x$ include continuity, absolute convergence of its defining series, $\sin 0 = 0$, and odd symmetry $\sin(-x) = -\sin x$. |
| Document | 18909 | 138 | **Power Series Expansion for Sine Function.** The sine function has the Taylor expansion $$\sin x = \sum_{n=0}^{\infty} (-1)^n \frac{x^{2n+1}}{(2n+1)!} = x - \frac{x^3}{3!} + \frac{x^5}{5!} - \cdots,$$ valid for all real $x$. |
| Document | 17171 | 325 | **Boundedness of $\sin x / x$.** For all real $x$, $$\left| \frac{\sin x}{x} \right| \le 1.$$ |
| Document | 11216 | 126 | **Limit of $x / \sin x$ at Zero.** $$\lim_{x \to 0} \frac{x}{\sin x} = 1.$$ |
| Document | 15227 | 2204 | **Derivative of Exponential at Zero.** Let $\exp x$ be the exponential function. Then $$\lim_{x \to 0} \frac{\exp(x) - 1}{x} = 1.$$ |
| Document | 15118 | 581 | **Derivative of Logarithm at One.** Let $\ln x$ be the natural logarithm for $x > 0$. Then $$\lim_{x \to 0} \frac{\ln(1 + x)}{x} = 1.$$ |
| **Answer** | 11592 | 2812 | **L'Hôpital's Rule.** Let $f$ and $g$ be real functions continuous on $[a, b]$ and differentiable on $(a, b)$, with $g'(x) \neq 0$ and $f(a) = g(a) = 0$. Then $$\lim_{x \to a^+} \frac{f(x)}{g(x)} = \lim_{x \to a^+} \frac{f'(x)}{g'(x)}.$$ |

Table 6: An example from TheoremQA-T

