# OpenReview forum: "Beyond Sequential Reranking: Reranker-Guided Search Improves Reasoning Intensive Retrieval"
_ICLR.cc/2026/Conference — ICLR 2026 Poster_

### Official Review · Reviewer_Y7xP · 2025-10-31

**Soundness:** 3
**Presentation:** 2
**Contribution:** 2
**Rating:** 6
**Confidence:** 3

**Summary:**

This paper proposes a novel method Reranker-Guided Search (RGS) to address two key limitations of the traditional “retrieval-reranking” pipeline: 1) Its performance is constrained by the quality of the Top-K documents returned by the initial retrieval stage; 2) The computational cost of large language model (LLM)-based rerankers is prohibitively high, limiting the number of documents that can be processed. RGS bypasses sequential reranking by performing a greedy search directly on the document proximity graph generated by an approximate nearest neighbor (ANN) algorithm. This allows for strategic prioritization of more promising documents for reranking based on the reranker's preferences. Experiments on BRIGHT, FollowIR, and M-BEIR demonstrate that RGS achieves significant performance gains over sequential reordering methods.

**Strengths:**

1.  Compared to the Retrieve-and-Rerank approach, reranking results are imperceptible to the retrieval process. In RGS, reranking results can help improve the retrieval process, thereby achieving performance optimization.
 2. On three challenging benchmarks: BRIGHT, FollowIR, and M-BEIR, RGS achieved significant performance improvements over both the traditional method and SlideGAR. This demonstrates its effectiveness.
 3. Experiments demonstrate that RGS is robust to noisy query embeddings. Even when query embeddings are of low quality, they can still identify relevant documents by leveraging the local structure of the document graph and guidance from the reordering module.

**Weaknesses:**

1.  RGS requires less computational effort, but each step's result depends on the previous one, resulting in longer processing delays compared to traditional reranking methods. In retrieval systems, this may be unacceptable to users. And RGS requires pre-constructing a proximity graph for the document collection. For large-scale document repositories, building and storing such a graph structure incurs additional preprocessing costs and memory overhead.
2. For an initial retrieval result that is completely irrelevant and isolated in the embedding space, RGS struggles to correct it through re-ranking into a completely different, highly relevant region. Does RGS possess a certain degree of corrective capability?
3. Although the effectiveness of RGS was validated across three datasets, these datasets are relatively small compared to real-world datasets such as MSMARCO. RGS requires validation of its effectiveness on a larger scale and should supplement its efficiency assessment.

**Questions:**

1. Given that RGS requires multiple steps to return its final search results, time delay must be considered as a key metric for evaluating RGS methods.
2. Case studies can be used to highlight improvements and enhancements to RGS, as well as whether it has a corrective effect on irrelevant retrieval results.

---

> ### Author Response · Authors · 2025-11-22
>
> Reply to W1: Regarding constructing a proximity graph
>
> We want to clarify that building a proximity graph is an essential step for all popular graph-based nearest neighbor search algorithms (HNSW, DiskANN, NSG). Therefore, our method adds literally **zero** preprocessing cost or memory. Actually, one of our insights is that rather than treating embedding search as a black box and only reranking its output, directly performing greedy search guided by the reranker model on the internal proximity graph is more efficient.
>
> Reply to W2: Corrective capability of RGS
>
> Good question. I think you are referring to our ablation studies in Figure 3. For the right plot, when the query embedding is pure noisy (perturbation level = 1), only our RGS method can achieve similar retrieval quality, exhibiting the corrective capability. For the left plot, when the doc embedding is pure noisy (perturbation level = 1), no algorithm can achieve meaningful recall. This is true because if the document embedding provides no useful information, then the only retrieval method is to linearly scan the corpus and pick the best documents you have seen. The recall is upper-bounded by the fraction of the corpus the linear scan is performed.
>
> Reply to W3: Thank you for your suggestion. The application of RGS is to cases where the embedding similarity between query and document can’t capture relevancy well. However, for traditional retrieval datasets, the embedding similarity of query and documents has already captured their relevancy very well (partially because the embeddings are trained on those datasets), therefore, using a reranker won’t produce much improvement. For example, using bge-reranker-v2 to rerank the top 100 documents retrieved from bge-en-v1.5-large improves the score from 42-48 (according to  [1]) for MSMARCO, where for BRIGHT, the baseline algorithm retrieve-and-rerank improves the score from 13-25 when reranking the top 100 documents. Second, for complicated retrieval tasks, it is by nature hard to create a large corpus dataset because of the labelling effort. We selected the largest datasets available to us; for example, the EDIS task from M-BEIR has a corpus of 1 million documents.
>
> Reply to Q1: Regarding running cost
>
> Thanks for raising this concern! We acknowledge that running time is important when measuring a retrieval method. We have reported the number of documents scanned in our main experiments, and also API calls, token usage, in Figure 7 in the Appendix. The use of LLM-based rerankers [2,3,4,5] has attracted increased attention in recent years. For LLM-based rerankers, especially those based on API calls, the exact running time depends on many different factors, like the current network traffic, how many requests the user is allowed to issue per minute. Thus, it’s more reliable to compare some environment-independent metrics like the number of documents scanned, API calls, and token usage.
>
>
> Reply to Q2: Case study.
>
> Thank you for your suggestion. We have added an example in the appendix to show that our RGS algorithm can lead to the correct answer even if the initial starting point is wrong and the cosine similarity doesn’t indicate the query document's relevance.
>
> [1] https://github.com/FlagOpen/FlagEmbedding/tree/master/research/llm_reranker
>
> [2] Sun, Weiwei, et al. "Is ChatGPT good at search? Investigating large language models as re-ranking agents." arXiv preprint arXiv:2304.09542 (2023).
>
> [3] Pradeep, Ronak, Sahel Sharifymoghaddam, and Jimmy Lin. "Rankvicuna: Zero-shot listwise document reranking with open-source large language models." arXiv preprint arXiv:2309.15088 (2023).
>
> [4] Qin, Zhen, et al. "Large language models are effective text rankers with pairwise ranking prompting." Findings of the Association for Computational Linguistics: NAACL 2024. 2024.
>
> [5] Weller, Orion, et al. "Rank1: Test-time compute for reranking in information retrieval." arXiv preprint arXiv:2502.18418 (2025).

---

### Official Review · Reviewer_BB8G · 2025-11-02

**Soundness:** 3
**Presentation:** 3
**Contribution:** 2
**Rating:** 4
**Confidence:** 3

**Summary:**

The proposed method is Reranker-Guided-Search (RGS), a novel technique that directly retrieves documents based on a reranker's preferences using a greedy search on a proximity graph, rather than following the traditional sequential retrieve-then-rerank pipeline.

**Strengths:**

-The proposed method aims at a critical retrieval bottleneck.

-For effective reranking with limited budget, RGS strategically selects the most promising documents to rerank, achieving significantly better performance (as evidenced by the reported gains of 3.5-5.1 points) without increasing computational cost.

-The method is empirically validated across multiple diverse benchmarks BRIGHT, FollowIR, M-BEIR, demonstrating its robustness and generalizability.

**Weaknesses:**

-The proposed method is highly dependent on the embedding model. Compared with Retrieve-and-Rerank in Figure2, the proposed method is less sensitive to the choice of embedding model with budget more than 200. However, the highly dependency on the embedding model is still similar to the Retrieve-and-Rerank method.

-Why the proposed method RGS is better for reasoning intensive search tasks? Which componets of RGS play a critical role for this advantage?

-RGS claimed that the novel introduction of documentation similarity will prioritize more promising documents and improving complexity query-document relationships. However, SlideGAR has used such information to improve search performance. The difference between them needs to be clear.

**Questions:**

-The paper focuses on the quality of initial top-k ranked results, and proposes a corresponding searching method to tackle the tradeoff between effectiveness and efficiency. The proposed metric rerank@k is used for evaluation in the experiments. But we need some traditional metric like ndcg@k for comparison in Table 1 and 2. Though NDCG@10 is also provided, the correlation between rerank@k and ndcg@k leaves unknown. Moreover, the reasonability of rerank@k is needed to be verified.

-The paper must be self-contained. Some notations like q_t\rightarrow c_i in Table 2 need to be explained.

-A detailed analysis should be provided to explain why the proposed method RGS performs less better on multi-modality retrieval datasets than scientific and math reasoning datasets.

-Time complexity of baseline methods retrieve-and-rerank and SlideGAR needs to be analyzed like section3.4.

---

> ### Author Response · Authors · 2025-11-22
>
> Reply to W1: On embedding model sensitivity
>
> Thanks for raising this concern! The point we want to convey through the ablation study in Figure 2 is that after applying our RGS on the graph to scan a certain number of documents, e.g., as you noted, 200 documents, the final retrieval accuracy is less sensitive to the initial document embedding. This is expected, as greedy search needs some steps to navigate in the graph towards the destination. Otherwise, the retrieval quality is affected by the starting point, which is determined by the quality of the initial embedding. As a comparison, the gap between different models for the retrieve-and-rerank baseline doesn’t seem to narrow even for the 500 budget.
>
> Reply to W2: Regarding why RGS is good at reasoning intensive tasks
>
> Our point is that RGS is good in any case where the embedding similarity between query and document doesn’t capture the relevance well. We believe that this applies to many challenging retrieval tasks, e.g. reasoning reasoning-intensive retrieval, multimodal retrieval, and instruction retrieval. Our RGS allows to use LLMs to judge relevance between query document pairs and adaptively select more promising documents in the corpus to send to the LLM reranker, and thus better than the standard retrieve-then-rerank.
>
> Reply to W3: Difference between RGS and SlideGAR
>
> Though both RGS (ours) and SlideGAR use an LLM reranker in the search process. There are fundamental differences in both the algorithm design and experimental results. Please see our comparison from line 153 to line 161. In short, SlideGAR still uses embedding scores to select documents (from the graph frontier) to send to LLMs, while we send all the expanded documents to the LLM reranker. This is a fundamental difference because the target scenario is that some document has very poor cosine similarity with the query but is actually relevant. In principle, RGS is able to find it, but SlideGAR may not. Please see their experimental comparison in Tables 1, 2, and 3.
>
> Reply to Q1: Regarding Rerank@k
>
> Sorry for the confusion. We want to clarify that “Reranker@k” is a resource constraint on the usage of the reranker rather than a metric. For example, we are proposing to measure: given a limited resource to rerank at most 100 documents, what is the best NDCG@10 a retrieval method can get? We are still reporting NDCG@10 in this case, which is a standard practice for BRIGHT. Note that in other papers (e.g., papers on the BRIGHT benchmark), it’s common to limit the reranker to rerank the top 100 documents, and here we are exploring this new dimension of reranker budget. We believe our scenario is practical, as most of the running time is spent on reranking.
>
> Reply to Q2: In Table 3, we apply the notations from the original M-BEIR paper. For each subtask, the query can be a text $q_t$, an image $q_i$, or a combination of both. Each candidate can be a text $c_t$, an image $c_i$, or a combination $(c_i, c_t)$. We have updated the notation in our paper (please see "dataset details"). Thanks for your suggestion.
>
> Reply to Q3: We believe that our experimental results show that our RGS algorithm performs well above the baseline algorithms on the multimodality dataset M-BEIR. Could you elaborate more about “less better on multi-modality retrieval datasets”.
>
> Reply to Q4: Time complexity for baseline methods
>
> Thanks for your suggestion! We have added them to our paper. Please section 3.4

---

### Official Review · Reviewer_zg7B · 2025-11-02

**Soundness:** 3
**Presentation:** 3
**Contribution:** 3
**Rating:** 4
**Confidence:** 4

**Summary:**

This paper proposes Reranker-Guided-Search (RGS), a graph-based retrieval pipeline that departs from the standard retrieve-then-sequential-rerank paradigm by using a greedy search on a proximity graph of document embeddings and letting an expensive LLM-based reranker steer which graph neighborhoods to expand. The method initializes from a small set of embedding-nearest seeds, iteratively expands the neighbors of high-ranked documents, and uses a sliding-window listwise reranker to keep the most promising candidates until a fixed reranker budget is exhausted. Empirically the authors evaluate RGS under a Reranker@𝑘 protocol on three challenging benchmarks, BRIGHT, FollowIR, and M-BEIR, and report consistent improvements over retrieve-and-rerank and a recent SlideGAR baseline, especially in low-reranker-budget regimes. The paper also presents ablations on embedding quality, perturbations to embeddings, and an analysis suggesting final performance is often limited by reranker alignment rather than search coverage.

**Strengths:**

The paper addresses an important practical bottleneck: how to get more value out of a small number of expensive reranker calls. The idea of combining graph-based ANNS traversal with reranker guidance is natural but underexplored, and the RGS design is a clean, well-motivated instantiation of that idea: it leverages the clustering hypothesis and the bi-metric intuition to trade cheap embedding similarity for targeted reranker effort. The experiments are broad and focused on modern, hard benchmarks that stress reasoning and multi-modality rather than only classical TREC-style tasks; showing gains across BRIGHT, FollowIR, and M-BEIR strengthens the claim of generality. The Reranker@𝑘 evaluation protocol is also a useful contribution because it isolates the real-world constraint of costly reranker usage and makes comparisons fairer.

**Weaknesses:**

First, practical costs and latency tradeoffs are not fully quantified: report wall-clock latency, token usage, and parallelization implications (RGS is sequential by design) so practitioners can judge deployment feasibility.
Second,  novelty relative to GAR/SlideGAR is claimed but the differences are somewhat qualitative.
Third, statistical significance and variance across random seeds or query subsets are not reported; adding confidence intervals would strengthen the empirical claims.

**Questions:**

See Weaknesses.

---

> ### Author Response · Authors · 2025-11-22
>
> Reply to W1: Regarding practical costs
>
> We have reported API calls, token usage in Figure 7 in the Appendix. For LLM-based rerankers, especially those based on API calls, the exact running time depends on many different factors, like the current network traffic, how many requests the user is allowed to issue per minute. Thus, it’s more reliable to compare some environment-independent metrics like the number of documents scanned, API calls, and token usage. For example, previous papers [1,2,3,4] fix the reranker to scan at most 100 documents and compare NDCG@10. We are doing the same in our main experiment.
>
> Reply to W2: Regarding the difference between ours and slideGAR
>
> From an algorithmic perspective, we believe that there are key differences between SlideGAR and our algorithm in how to explore the graphs. Please see our discussion section 3.2, which explains the performance gap between our method and the SlideGAR baseline.
>
> Prior work shows search-based reranking on standard retrieval sets such as TREC. Our work shows the potential of graph-based reranking on reasoning retrieval tasks. In these tasks, dense retrievers struggle because query–document similarity does not reliably surface the documents needed for multi-step reasoning. This is where graph-based and search-based reranking becomes useful. We also construct a combined evaluation suite from existing datasets to measure performance on reasoning retrieval in a consistent way.
>
> Reply to W3: Statistical Significance Analysis
>
> For BRIGHT, we evaluate all 1,386 queries in the test set. For FollowIR, we evaluate all 349 queries in the test set. For M-BEIR, we follow sample 100 queries from each subtask, resulting in a total of 1,600 queries.
>
> We now conduct a statistical significance analysis on the M-BEIR evaluation. Since NDCG@10 is bounded in [0,1], Popoviciu’s inequality implies that its per-query variance is at most $(1-0)^2/4 = 1/4$, i.e., the standard deviation is at most 0.5. With 1,600 queries, this gives a worst-case standard error of: $SE = \frac{0.5}{\sqrt{1600}} = 0.0125$. Our observed improvement on M-BEIR is 0.082 under Reranker@500 in NDCG@10. Under the worst-case variance assumption, this corresponds to the Z-score: $Z = \frac{0.082}{0.0125} = 6.56$ which yields a two-sided p-value of approximately: $p\approx 2.7*10^{-11}$ Thus, even under the most conservative possible assumptions about the variance of per-query NDCG@10, the improvement is statistically significant by a very large margin.
>
> [1] Sun, Weiwei, et al. "Is ChatGPT good at search? Investigating large language models as re-ranking agents." arXiv preprint arXiv:2304.09542 (2023).
>
> [2] Pradeep, Ronak, Sahel Sharifymoghaddam, and Jimmy Lin. "Rankvicuna: Zero-shot listwise document reranking with open-source large language models." arXiv preprint arXiv:2309.15088 (2023).
>
> [3] Qin, Zhen, et al. "Large language models are effective text rankers with pairwise ranking prompting." Findings of the Association for Computational Linguistics: NAACL 2024. 2024.
>
> [4] Weller, Orion, et al. "Rank1: Test-time compute for reranking in information retrieval." arXiv preprint arXiv:2502.18418 (2025).

---

### Official Review · Reviewer_nmG8 · 2025-11-02

**Soundness:** 4
**Presentation:** 3
**Contribution:** 4
**Rating:** 8
**Confidence:** 5

**Summary:**

The paper presents reranker guided search (RGS). The technique is summarized in Algorithm 1. Essentially, the approach iteratively decides which documents to retrieve next by (1) reranking the current set of retrieved docs, (2) using the top ranked doc (after reranking) to then decide which new docs to include, and repeating at step (1) until the budget is fulfilled. The results on multiple text and multi-modal datasets are strong.

**Strengths:**

S1. Retrieve and rerank is becoming dated. Listwise is certainly an improvement over cross-encoder, but the design space for search is massive yet very under-explored for papers that report results on the top datasets, including the ones mentioned in this paper. It is great to see new creative works like RGS being applied, and with seemingly good results. That being said, it's hard to calibrate the strength of the results without more baselines.

S2. The paper includes extensive results and thoughtful analysis. I include some questions about new analysis, and questions about baselines, but the work is already very informative without these.

S3. The results with perturbation are particularly fascinating. The field at large has started to question the need for vector search given that reranking can be used to filter down an initial candidate set, and this is a result in that same spirit.

**Weaknesses:**

W1. It is a major weakness that ReasonIR is not included as a baseline, nor other reranker alternatives like JudgeRank or Rank-R1. Basically, the results in Table 1-3 are very limited in their baselines, only using BGE embedding and not looking at many rerank alternatives.

W2. Listwise rerankers are know to be very sensitive to order, for example see https://arxiv.org/abs/2306.17563 which shows that listwise rerankers in reverse or when re-run multiple times can have very different results. For this reason, it would be very helpful to show Recall@R in Tables 1-3. We can not tell otherwise if it is recall that is helping or a different ordering of results fed to the reranker. The text has "the number [recall] is only 31% on BRIGHT [at top-100]", which does suggest RGS is helping with recall, but it's not clear which datasets this holds for. Using Recall@R can even be used to compare all the methods without applying any reranking.

W3. Intuitively, it's not clear why RGS should help with instruction following.

W4. The benefits of RGS decline rapidly after budget hits 500. I actually wonder if it will start to get worse as the budget increases further. More importantly, it's not clear so clear whether 100 budget is needed, or much of the benefit can be achieved with only 50 or even 20.

W5. Although time complexity of RGS is reported, there is no cost analysis between the different methods.

**Questions:**

How does Recall@R compare using embedding-only vs embedding + proximity graph? Basically, what is the recall before listwise reranker applied?

"We believe that these benchmarks represent the most complicated retrieval tasks available, which can’t be solved by simple semantic similarity, and serve as a better testbed for evaluating the efficiency of different high-performing retrieval methods." There are a range of other tasks worth considering beyond this set. CRUMB (https://arxiv.org/abs/2509.07253) provides a good overview.

Is it necessary to run the reranker each time at step 12 in the algo? Could the reranking simply be done at the end?

---

> ### Author Response · Authors · 2025-11-22
>
> Reply to W1: Thank you for the suggestions. We evaluated several existing reranker models (e.g., bge-reranker, voyage-rerank) during the early stage of this work and observed that their performance gains plateau—or even decrease—once the number of scanned documents exceeds a few dozen. This behavior is also noted in [1]. Motivated by this limitation, we subsequently explored the use of general LLMs as rerankers.
>
> In our experiments and ablation studies (Figure 2 and Table 5), we evaluated the reranker models listed below. Due to computational and budget constraints, ablation studies were performed on a subset of datasets.
>
> Embedding Models: BGE-micro, BGE-base, BGE-Large, SFR-Mistral
>
> Reranker Models: Gemini-2.0-Flash, GPT-4.1-mini
>
> Reply to W2 & Q1: Thanks for your question! This is an interesting question. We report NDCG@10 because it is the standard metric used on the BRIGHT benchmark [4]. Additionally, we notice that most of the datasets have some test data with >10 groundtruth documents (e.g. one test data from economics has 85 groundtruth docs), which means that no model can reach perfect Recall@10. It’s interesting to differentiate whether the improvement comes from better reranking the order or better exploration of the corpus. We have conducted an ablation study in Figure 4. For each query and different reranker budgets, we define the following variables
>
> GT_scanned = the number of groundtruth documents scanned by the reranker during the retrieval process
>
> GT_retrieved = the number of groundtruth documents returned within the top 10
>
> GT_count = the number of groundtruth documents for this query.
>
> We report:
>
> Fraction retrieved: min(GT_retrieved,10) / min(GT_count,10) as orange
>
> Fraction scanned: min(GT_scanned,10) / min(GT_count,10) as orange + light orange
>
> Note that the fraction we report is different from the standard Recall@10 metric; we perform this informal normalization so that the maximal possible score is 1.
>
> Our results show that as the reranker budget increases, the orange part grows slowly, but the light orange fraction is constantly increasing. This provides evidence that our RGS does find more groundtruth documents than other algorithms, but this may not be reflected in the final recall or NDCG, because of various reasons, e.g., LLM preference doesn’t align with human labels, listwise reranker is sensitive to order (as you mentioned). For reference, for the biology dataset, the “Fraction retrieved” is 0.151 in Figure 4 if the reranker is not applied at all. This can be conceived as an approximation for the “Recall@10” metric you requested.
>
>
> Reply to W3: Reasoning-intensive tasks are challenging since retrieval based on semantic similarity doesn't work well. We hypothesize that RGS should apply as long as this condition is met. We choose instruction following retrieval because one document may answer one query but not another one with a slightly different intent, which is challenging for methods simply relying on embedding similarity to judge relevance.
>
> Reply to W4: Most results on the BRIGHT benchmark (e.g. [2], [3]) rerank the first 100 documents. We let our RGS scan the same number of documents. Please see our Figure 4, the number of groundtruth documents keeps increasing as the budget increases, but the improvement may not be reflected in NDCG@10.
>
> Reply to W5: In our experiments, most of the time is spent on calling Gemini APIs and waiting for its feedback. The response latency of Gemini varies daily due to different network situations, which we believe is independent of our research. That’s why we focus on reporting the number of documents scanned, and also token usage and API calls (see Figure 5 in the Appendix)
>
> Reply to Q2: Thank you for your reference! We will experiment with that in our future work.
>
> Reply to Q3: We believe you are talking about step 11. Now the LLMs have a limitation on attention context, which means that their output quality will decrease if we ask them to rerank too many documents at the same time. Given an ordered list of candidate documents, if a new document comes, we first attach the new document at the end, and then do a “sliding window” to move it to the front. This is necessary as we have to maintain an up-to-date ordering of the candidate documents. We agree that this may not be optimal, and it is worth studying what the optimal way to combine LLM rerankers and greedy search is.
>
> [1]  Jacob, Mathew, et al. "Drowning in documents: consequences of scaling reranker inference." arXiv preprint arXiv:2411.11767 (2024).
>
> [2] https://github.com/VectorSpaceLab/agentic-search/tree/main/BGE-Reasoner
>
> [3] Long, Meixiu, et al. "Diver: A multi-stage approach for reasoning-intensive information retrieval." arXiv preprint arXiv:2508.07995 (2025).
>
> [4] Su, Hongjin, et al. "Bright: A realistic and challenging benchmark for reasoning-intensive retrieval." arXiv preprint arXiv:2407.12883 (2024).

---

### Meta-Review · Area_Chair_tPtM · 2026-01-09

**Summary:**

This paper proposes Reranker-Guided-Search (RGS), a graph-based retrieval pipeline, which uses a greedy search on a proximity graph of document embeddings and letting an expensive LLM-based reranker steer which graph neighborhoods to expand. The proposed method is simple and effective on the benchmarks. Reviewers' concerns include:
1. The proposed method is highly dependent on the embedding model
2. SlideGAR has used similar ideas.

Overall, the proposed method is very simple, which can be described with 10 lines of pseudocode. So reviewers have some concerns on the novelty. Considering the simplicity and effectiveness, I am slightly leaning towards acceptance.

**Reviewer Concerns:**

Concerns are mostly addressed. But whether it has enough of novelty is somewhat a matter of taste.

**Reviewer Scores:**

8,4,4,6

---

### Decision · Program_Chairs · 2026-01-26

Accept (Poster)